# Structural basis of nucleoside and nucleoside drug selectivity by concentrative nucleoside transporters

**Zachary Lee Johnson[1], Jun-Ho Lee[1], Kiyoun Lee[2], Minhee Lee[2], Do-Yeon Kwon[2], Jiyong Hong[2,3], Seok-Yong Lee[1]\***

[1]Department of Biochemistry, Duke University Medical Center, Durham, United States; [2]Department of Chemistry, Duke University, Durham, United States; [3]Department of Pharmacology and Cancer Biology, Duke University Medical Center, Durham, United States

**Abstract** Concentrative nucleoside transporters (CNTs) are responsible for cellular entry of nucleosides, which serve as precursors to nucleic acids and act as signaling molecules. CNTs also play a crucial role in the uptake of nucleoside-derived drugs, including anticancer and antiviral agents. Understanding how CNTs recognize and import their substrates could not only lead to a better understanding of nucleoside-related biological processes but also the design of nucleoside-derived drugs that can better reach their targets. Here, we present a combination of X-ray crystallographic and equilibrium-binding studies probing the molecular origins of nucleoside and nucleoside drug selectivity of a CNT from *Vibrio cholerae*. We then used this information in chemically modifying an anticancer drug so that it is better transported by and selective for a single human CNT subtype. This work provides proof of principle for utilizing transporter structural and functional information for the design of compounds that enter cells more efficiently and selectively.

*For correspondence: sylee@biochem.duke.edu

**Competing interests:** The authors declare that no competing interests exist.

## Introduction

Nucleosides play critical roles in biology as precursors to nucleic acids and the energy currency of the cell and also serve as signaling molecules (*King et al., 2006*; *Rose and Coe, 2008*; *Molina-Arcas et al., 2009*). Furthermore, nucleoside analogs have clinical applications as anticancer and antiviral drugs (*Damaraju et al., 2003*; *Jordheim and Dumontet, 2007*). Because of their immense biological and clinical importance, efficient entry of nucleosides and their analogs into the cell is crucial to human health and disease. Cellular entry is accomplished by a class of membrane proteins known as nucleoside transporters (NTs). There are two types of NTs in humans: concentrative nucleoside transporters (CNTs) and equilibrative nucleoside transporters (ENTs). CNTs utilize the energy of ion gradients to actively transport nucleosides into the cell against their concentration gradients while ENTs transport nucleosides down their chemical gradients without the requirement of any additional energy source (*Gray et al., 2004b*).

In addition to nucleosides, NTs are responsible for the transport of a wide range of nucleoside-derived anticancer (e.g., gemcitabine and 5-fluorouridine) and antiviral (e.g., ribavirin) drugs (*Farre et al., 2004*; *Marechal et al., 2009*; *Rabascio et al., 2010*; *Bhutia et al., 2011*; *Doehring et al., 2011*; *Fukao et al., 2011*; *Rau et al., 2013*). Both NT families possess subtype-dependent nucleoside specificities and tissue distributions, while CNTs are more highly subtype-specific for their substrates and distributions than ENTs (*Gray et al., 2004b*; *Paproski et al., 2013*). As a result, different NT subtypes are responsible for the transport of different types of nucleosides and nucleoside drugs, and expression levels of different NTs can predict how patients with certain types of cancer and viral infection will

**eLife digest** DNA molecules are made from four bases—often named 'G', 'A', 'C', and 'T'—that are arranged along a backbone made of sugars and phosphate groups. Chemicals called nucleosides are essentially the same as these four building blocks of DNA (and other similar molecules) but without the phosphate groups.

Proteins called nucleoside transporters are found in the membranes that surround cells and can pump nucleosides into the cell. These transporters also allow drugs that are made from modified nucleosides to enter cells; however, it was previously unclear how different transporters recognized and imported specific nucleosides.

Like other proteins, nucleoside transporters are basically strings of amino acids that have folded into a specific three-dimensional shape. A protein's shape is often important for defining what that protein can do, as often other molecules must bind to proteins—much like a key fitting into a lock. Johnson et al. have now revealed the three-dimensional structure of one nucleoside transporter protein bound to different nucleosides and nucleoside-derived chemicals, including three anti-cancer drugs and one anti-viral drug. Some of these chemicals were shown to bind more strongly to the transporter protein than others, and examining the three-dimensional structures revealed that the different chemicals interacted with slightly different amino acids in the transporter protein.

Johnson et al. then used this information to chemically modify an anticancer drug so that it is transported more easily into cells and is imported by only one of the subtypes of nucleoside transporters that are found in humans. This provides proof of principle that information about the structure and function of a transporter protein can help to redesign chemicals such that they can enter cells more efficiently, and to tailor them for transport by specific transporters. A similar approach may in the future allow researchers to design new nucleoside-derived drugs that are better at getting inside specific cells and, as such, provide effective treatments against cancers and viral infections.

respond to nucleoside-drug treatment (*Mackey et al., 1998*; *Farre et al., 2004*; *Spratlin et al., 2004*; *Gray et al., 2004a*; *Damaraju et al., 2009*; *Marechal et al., 2009*; *Rabascio et al., 2010*; *Bhutia et al., 2011*; *Doehring et al., 2011*; *Fukao et al., 2011*; *Rau et al., 2013*). Since there are three different isoforms of human CNTs (hCNT1-3) that possess differing nucleoside and nucleoside-drug specificities and tissue distributions, greater knowledge of the molecular origins of nucleoside selectivity by CNTs could potentially lead to better-tailored nucleoside drug delivery as well as a better understanding of CNT-mediated physiological processes.

CNTs belong to the solute carrier (SLC) superfamily, constituting the family SLC28. The SLC super-family, composed of 52 families, is responsible for the transport of ions, metabolites, neurotransmitters, and drugs in humans. Several SLC families are of particular clinical interest because of their roles in drug absorption, distribution, metabolism, and excretion (ADME) (*Schlessinger et al., 2013*). The recent determination of the structures of several SLC transporters has advanced our understanding of the inner workings of these transporters and expanded the applicability of structure-based ligand discovery using computational methods (*Gao et al., 2009*; *He et al., 2010*; *Hu et al., 2011*; *Newstead et al., 2011*; *Johnson et al., 2012*; *Pedersen et al., 2013*). Although this computational method of ligand discovery is a valuable approach, it cannot accurately predict the energetically important interactions between ligands and transporters, and therefore experimental approaches should be pursued to understand the principles of ligand and drug selectivity by these transporters.

The crystal structure of a CNT from *Vibrio cholerae* (vcCNT) presents the first opportunity to examine specific nucleoside recognition by CNTs from a structural perspective (*Johnson et al., 2012*). vcCNT is an excellent model system to study hCNTs: it utilizes a $Na^+$ gradient for nucleoside transport like hCNTs and shares high sequence identity (36–39%) with hCNTs with particularly high sequence identity for the nucleoside-binding site (64% with hCNT1, 73% with hCNT2, 91% with hCNT3). For these reasons, vcCNT has been identified as an optimal candidate for structure-based ligand discovery using computational methods (*Schlessinger et al., 2013*).

Here, we have exploited a combination of X-ray crystallographic studies and equilibrium-binding measurements of vcCNT to understand the structural basis of CNT selectivity. We have discovered that

CNTs use a unique mode of nucleoside recognition that is suitable for its function as a transporter. Using the insights gained from these studies, we have chemically modified the anticancer drug gemcitabine and found that its binding affinity for vcCNT is greatly enhanced. Furthermore, the modified compound now possesses subtype-specific transport among human CNTs. Follow-up structural and mutational studies revealed the origin of subtype-specificity of the modified compound. Not only do our studies illuminate the structural basis of nucleoside selectivity by CNTs but they also provide proof of principle for utilizing membrane transporter structures for the design of drugs with more selective delivery (*Han and Amidon, 2000*; *Majumdar et al., 2004*).

## Results

### The nucleoside-binding site of vcCNT-7C8C and equilibrium-binding measurements

vcCNT forms a homotrimer with each protomer possessing its own nucleoside-binding site and permeation pathway (*Figure 1A*). The protomer adopts a new fold and is divided into two domains: the scaffold domain that is responsible for trimerization and maintaining the overall architecture of the transporter (light blue, *Figure 1B*), and the transport domain where nucleoside binding and transport occur (other colors, *Figure 1B*). The nucleoside-binding site, facing the trimer axis, is formed at the center of the transport domain between the tips of two helical hairpins (HP1 and HP2) and two partially unwound transmembrane helices (TM4 and TM7) (*Johnson et al., 2012*).

We solved the structure of a uridine-bound double mutant of vcCNT (Leu 7 to Cys and Ile 8 to Cys, termed 7C8C) at 2.1 Å, which is higher resolution than the 2.4-Å wild-type structure. This mutant was originally designed to introduce binding sites for hydrophobic mercury compounds for heavy-atom phasing. The high-resolution mutant structure revealed another water in the binding site that bridges the 4-carbonyl of the uracil base with Glu 156 but was otherwise identical to the wild-type structure (*Figure 1C*, *Figure 1—figure supplement 2*; *Table 1*). The mutations do not affect transporter function significantly (*Figure 1—figure supplement 1*).

The structure of vcCNT bound to uridine revealed that the interactions can be divided into two groups: those that involve the ribose moiety and those with the nitrogenous base (*Figure 1C*). To determine the energetic contributions of each of the interactions between nucleoside and vcCNT, we developed a fluorescence-anisotropy-based competition assay for measuring the equilibrium dissociation constants ($K_D$s) for a variety of nucleosides and nucleoside analogs using the fluorescent cytidine analog pyrrolo-cytidine (*Table 2*; *Damaraju et al., 2011*). We calculated the $K_D$ for uridine to be 36 µM (*Figure 1D*), which is similar to the reported $K_m$ values of uridine for hCNTs ($K_m$ = 22–80 µM), further suggesting that vcCNT is a good model system to study hCNTs (*Molina-Arcas et al., 2009*).

### Nucleobase interactions

Human CNTs have differing nucleoside-base preferences: hCNT1 mainly transports pyrimidines, hCNT2 prefers purines, and hCNT3 is broadly selective for both pyrimidines and purines (*Gray et al., 2004b*; *Molina-Arcas et al., 2009*). The uracil base interacts with residues on HP1 (Gln 154 directly and Thr 155 and Glu 156 through water molecules) and TM4 (Val 188 via van der Waals interactions). To examine the energetics of these interactions, we measured $K_D$s of vcCNT for uridine analogs with modifications to the uracil base as well as other nucleosides.

The anticancer drug zebularine is a uridine analog with no substituent at the C4 position of the pyrimidine base. Zebularine exhibits a ~threefold loss of binding affinity ($K_D$ = 120 µM) relative to uridine. To deduce the structural basis of the reduced binding affinity, we solved the crystal structure of vcCNT bound to zebularine (*Figure 2A*, *Figure 2—figure supplement 1*). The crystal structure shows that the side chain of Glu 156 adopts a different rotamer position probably because it is unable to form the water-mediated interaction with the C4-carbonyl of the uracil base, consistent with the loss of binding affinity.

N3 of the uracil base interacts with both Thr 155 and Glu 156 through a single water molecule that is coordinated by both residues. We measured the affinity of vcCNT for 3-methyluridine, which contains a methyl group at this position that blocks the water-mediated interaction, and we found that it significantly decreased the binding affinity ($K_D$ = 520 µM, *Figure 2B*, *Figure 2—figure supplement 2*). The purine nucleoside adenosine and the antiviral guanosine analog ribavirin also possess similarly weaker binding affinity ($K_D$ = 470 and 1530 µM, respectively). To examine the structural basis of the

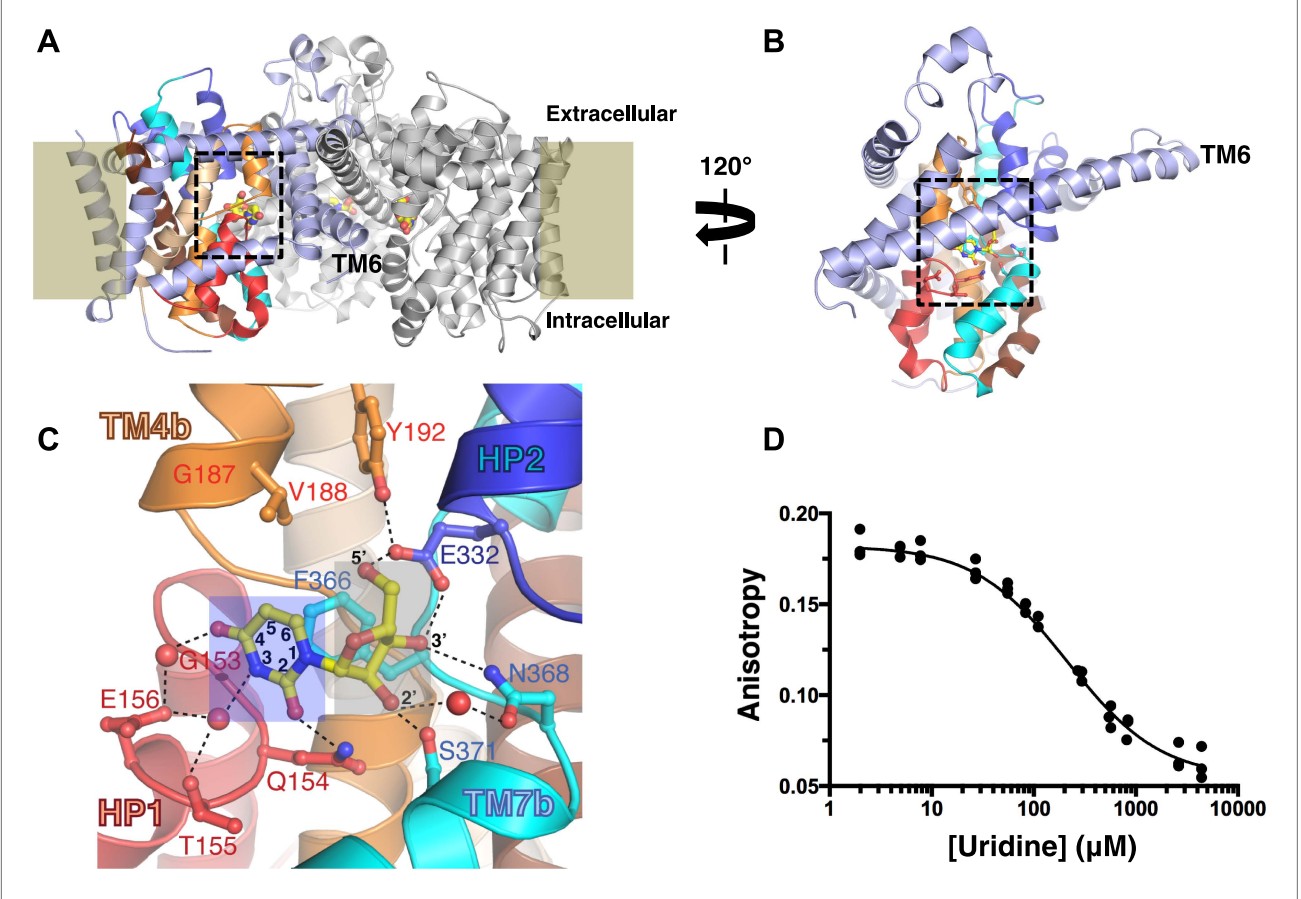

**Figure 1**. The nucleoside-binding site of vcCNT and fluorescence-anisotropy-based competition assay. (**A**) The vcCNT-7C8C trimer viewed from within the plane of the membrane. The location of the membrane is marked by rectangles. The scaffold domain of one protomer is colored light blue, and the transport domain is colored red, blue, orange, cyan, wheat, and brown. The other two protomers are colored gray. Uridine is shown bound to each protomer in stick representation. The nucleoside-binding site is delineated with dashed lines. vcCNT-7C8C functions similarly to wild type (**Figure 1—figure supplement 1**). (**B**) The vcCNT-7C8C protomer. The structure is rotated 120° about the trimer axis relative to **A**, zoomed in, and the other two protomers have been removed for clarity. (**C**) Nucleoside-binding site. Amino acid residues that interact with the uridine are labeled and shown in stick representation and were used for sequence identity calculation with hCNTs. Hydrogen bonds are shown as dashed lines. The uracil base is marked with a blue box, and the ribose is marked with a gray box. For a stereo view of the electron density in the nucleoside-binding site, see **Figure 1—figure supplement 2**. (**D**) Fluorescence titration of vcCNT with uridine. Uridine was titrated into solution containing vcCNT and the fluorescent nucleoside pyrrolo-cytidine, anisotropy was measured, and data were fit to a single-site competitive binding model to obtain a $K_D$ of 36 ± 3 μM (mean ± SEM, n = 3 measurements).

The following figure supplements are available for figure 1:

**Figure supplement 1**. vcCNT-7C8C maintains nucleoside transport activity.

**Figure supplement 2**. Electron density at the nucleoside-binding site of vcCNT-7C8C-uridine.

reduced affinity, we solved the crystal structures of vcCNT-7C8C bound to adenosine (**Figure 2C**, **Figure 2—figure supplement 3**) and ribavirin (**Figure 2D**, **Figure 2—figure supplement 4**). The structures reveal that the bulky purine base displaces the water observed in the uridine structure while maintaining similar modes of ribose binding, corroborating the idea that the reduced affinities of adenosine and ribavirin are due to the loss of the water-mediated interactions and highlighting the importance of the water coordinated by Thr 155 and Glu 156 in nucleoside recognition by CNTs.

To test the energetic contribution of substituents at the C5 position of the pyrimidine ring, we measured the $K_D$ for the anticancer drug 5-fluorouridine to be 16 μM, which is ~twofold lower than the $K_D$ for uridine. We also solved the crystal structure of vcCNT-7C8C bound to 5-fluorouridine (**Figure 2E**).

**Table 1.** Data collection and refinement statistics

| | vcCNT-7C8C-uridine | vcCNT-zebularine | vcCNT-7C8C-adenosine | vcCNT-7C8C-ribavirin |
|---|---|---|---|---|
| Data collection | | | | |
| Space group | P6$_3$ | P6$_3$ | P6$_3$ | P6$_3$ |
| Cell dimensions | | | | |
| a, b, c (Å) | 119.7, 119.7, 83.1 | 119.8, 119.8, 82.7 | 120.0, 120.0, 83.5 | 119.7, 119.7, 83.6 |
| α, β, γ (°) | 90, 90, 120 | 90, 90, 120 | 90, 90, 120 | 90, 90, 120 |
| Resolution (Å) | 2.08 (2.12–2.08)* | 2.90 (2.95–2.90) | 3.10 (3.15–3.10) | 2.80 (2.85–2.80) |
| $R_{sym}$ or $R_{merge}$ | 0.052 (0.554) | 0.141 (0.766) | 0.104 (0.567) | 0.114 (0.758) |
| $I/\sigma I$ | 42.0 (2.3) | 13.5 (1.9) | 14.1 (1.3) | 20.2 (1.8) |
| Completeness (%) | 99.3 (93.0) | 99.9 (100.0) | 99.0 (96.5) | 99.8 (100.0) |
| Redundancy | 5.9 (4.3) | 5.8 (5.1) | 4.2 (3.4) | 7.0 (6.3) |
| Refinement | | | | |
| Resolution (Å) | 2.08 (2.13–2.08) | 2.91 (3.13–2.91) | 3.10 (3.41–3.10) | 2.80 (2.98–2.80) |
| No. reflections | 40368 (2381) | 14932 (2798) | 12429 (2887) | 16840 (2635) |
| $R_{work}/R_{free}$ (%) | 20.2/23.3 | 21.1/24.7 | 22.4/26.9 | 22.2/25.9 |
| No. atoms | | | | |
| Protein | 2834 | 2868 | 2925 | 2947 |
| Ligand/ion | 17/1 | 16/1 | 19/1 | 17/1 |
| Water/detergent | 130/33 | 3/33 | 3/33 | 2/33 |
| B-factors | | | | |
| Protein | 44.1 | 45.6 | 72.1 | 58.7 |
| Ligand/ion | 34.2/31.3 | 25.6/47.6 | 61.9/71.0 | 54.2/64.5 |
| Water/detergent | 54.4/62.6 | 29.3/56.4 | 54.6/89.0 | 48.1/72.3 |
| R.m.s deviations | | | | |
| Bond lengths (Å) | 0.005 | 0.003 | 0.002 | 0.003 |
| Bond angles (°) | 0.782 | 0.676 | 0.617 | 0.709 |
| | **vcCNT-7C8C-5-fluorouridine** | **vcCNT-7C8C cytidine** | **vcCNT-7C8C-pyrrolo-cytidine** | **vcCNT-7C8C-gemcitabine** |
| Data collection | | | | |
| Space group | P6$_3$ | P6$_3$ | P6$_3$ | P6$_3$ |
| Cell dimensions | | | | |
| a, b, c (Å) | 119.8, 119.8, 83.2 | 120.0, 120.0, 82.5 | 119.6, 119.6, 83.1 | 119.0, 119.0, 82.3 |
| α, β, γ (°) | 90, 90, 120 | 90, 90, 120 | 90, 90, 120 | 90, 90, 120 |
| Resolution (Å) | 2.30 (2.34–2.30) | 2.60 (2.64–2.60) | 2.75 (2.80–2.75) | 2.90 (2.97–2.90) |
| $R_{sym}$ or $R_{merge}$ | 0.067 (0.500) | 0.094 (0.665) | 0.083 (0.656) | 0.061 (0.554) |
| $I/\sigma I$ | 28.0 (1.6) | 24.0 (2.0) | 22.6 (1.9) | 30.8 (1.9) |
| Completeness (%) | 97.3 (79.2) | 99.3 (98.7) | 99.9 (100.0) | 99.6 (99.0) |
| Redundancy | 4.0 (2.4) | 6.4 (6.0) | 5.8 (5.5) | 5.7 (4.6) |
| Refinement | | | | |
| Resolution (Å) | 2.30 (2.38–2.30) | 2.61 (2.75–2.61) | 2.75 (2.92–2.75) | 2.91 (3.13–2.91) |
| No. reflections | 29380 (2089) | 20600 (2754) | 17698 (2784) | 14682 (2751) |
| $R_{work}/R_{free}$ (%) | 20.0/21.6 | 21.4/23.9 | 20.2/24.7 | 22.7/25.8 |
| No. atoms | | | | |
| Protein | 2937 | 2898 | 2921 | 2883 |

*Table 1. Continued on next page*

*Table 1. Continued*

| | vcCNT-7C8C-5-fluorouridine | vcCNT-7C8C cytidine | vcCNT-7C8C-pyrrolo-cytidine | vcCNT-7C8C-gemcitabine |
|---|---|---|---|---|
| Ligand/ion | 18/1 | 17/1 | 20/1 | 18/1 |
| Water/detergent | 58/33 | 28/33 | 3/33 | 9/33 |
| B-factors | | | | |
| Protein | 54.4 | 59.1 | 60.9 | 76.7 |
| Ligand/ion | 41.4/47.2 | 46.1/48.6 | 48.4/54.1 | 59.6/70.1 |
| Water/detergent | 59.1/67.9 | 60.3/62.6 | 47.9/72.7 | 58.8/101.4 |
| R.m.s deviations | | | | |
| Bond lengths (Å) | 0.002 | 0.002 | 0.003 | 0.004 |
| Bond angles (°) | 0.660 | 0.609 | 0.646 | 0.664 |

*Highest resolution shell is shown in parenthesis.

The structure shows that the modified uracil base still fits into the nucleoside-binding site without any structural rearrangements of the protein. To understand the origin of enhanced affinity by the addition of fluorine at the C5 position, we compared the binding affinity of 5-fluorouridine to other C5-substituted uridine analogs with differing electronegativities and atomic radii (*Table 3*). Fluorine is highly electronegative (3.98 on the Pauling scale) and relatively small (1.47 Å radius). When another highly electronegative but slightly larger halogen, chlorine (3.16, 1.75 Å), is substituted at the C5 position, a similar $K_D$ is observed (14 µM). However, when the large, weakly electronegative substituent iodine (2.66, 1.98 Å) or a methyl group (2.55, 2.00 Å) is added, we observe an increase in $K_D$ (~60 µM). In short, we observed an increase in affinity with smaller, highly electronegative substituents but a decrease in affinity with larger, less electronegative substituents. What structural feature could account for the differing affinities of these compounds?

These differences may result from interactions of the nucleosides with Phe 366. In the structure of vcCNT-7C8C bound to uridine, Phe 366 appears to interact with both the uracil base and the ribose of the nucleoside. Phe 366 forms an offset π–π interaction with the aromatic pyrimidine ring and also forms CH–π interactions with the ribose (*Figure 1C*, *Figure 3A,B*). The addition of a small, highly electronegative substituent at the C5 position could strengthen the interaction between the pyrimidine ring and Phe 366, as is the case with many π–π interactions (*Hunter and Sanders, 1990*; *Ringer et al., 2006*).

Notably, Phe 366 is universally conserved between members of the CNT family, and its functional importance has never been tested. To examine the role of Phe 366 in nucleoside recognition, we mutated this residue and performed isothermal titration calorimetry (ITC) experiments. We found that the requirement for Phe at this position is strict, as mutation of this residue to alanine or tyrosine or tryptophan results in significantly decreased binding affinity for uridine while not affecting the stability significantly (*Figure 3C–F*, *Figure 3—figure supplement 1*). Therefore, we suggest that Phe 366 plays a critical role in recognition of the nucleoside by CNTs.

## Ribose interactions

Amino acid residues that interact with the ribose in the vcCNT structure (Glu 332, Asn 368, and Ser 371) are invariant between hCNTs and vcCNT. To probe these interactions, we measured $K_D$s for deoxyuridines from which each of the ribose hydroxyls have been removed. Both 3'- and 5'-deoxyuridine yielded $K_D$ values greater than 2800 µM (*Figure 4A*), while the binding affinity for 2'-deoxyuridine is not as drastically affected ($K_D$ = 170 µM). Several anticancer nucleoside analog drugs contain modifications at the C2' position of the ribose. Gemcitabine, for example, is a cytidine analog with two fluorine atoms bonded to C2'. The measured $K_D$ of gemcitabine for vcCNT ($K_D$ = 1370 µM) is ~22-fold higher than that for cytidine ($K_D$ = 61 µM) (*Figure 4B*, *Figure 4—figure supplement 1*). To understand the structural basis of this significant reduction in binding affinity associated with the fluorine substitutions, we solved the crystal structure of vcCNT-7C8C bound to gemcitabine (*Figure 4C*, *Figure 4—figure supplement 2*). In the uridine-bound structure, a CH–π interaction was observed between the

**Table 2.** K$_D$ values for nucleosides and nucleoside analog drugs calculated from fluorescence titrations

| Compound | K$_D$ (µM)* |
|---|---|
| uridine | 36 ± 3 |
| cytidine | 61 ± 5 |
| adenosine | 470 ± 100 |
| gemcitabine | 1,370 ± 430 |
| ribavirin | 1,530 ± 350 |
| zebularine | 120 ± 5 |
| 3-methyluridine | 520 ± 80 |
| 5-fluorouridine | 16 ± 1 |
| 5-chlorouridine | 14 ± 1 |
| 5-iodouridine | 58 ± 5 |
| 5-methyluridine | 61 ± 7 |
| 2'-deoxyuridine | 170 ± 10 |
| 3'-deoxyuridine | >2,800 |
| 5'-deoxyuridine | >2,800 |
| cytarabine | >3,000 |
| pyrrolo-cytidine | 0.94 ± 0.17 |
| pyrrolo-gemcitabine | 23.5 ± 0.2 |

*Titrations were performed in triplicate and data were fit globally. All values are given as means ± SEM. See **Table 2—source data 1** for fluorescence data used in calculating K$_D$ values.

**Source data 1**. Fluorescence data for K$_D$ calculations. For pyrrolo-cytidine and pyrrolo-gemcitabine, individual solutions with fixed nucleoside concentrations and increasing concentrations of vcCNT were prepared, the fluorescence anisotropy was measured, and the data were fit globally to a single-site binding model accounting for ligand depletion. For all other nucleosides, the nucleoside of interest was titrated into solution containing vcCNT and pyrrolo-cytidine 5 µl at a time, the fluorescence anisotropy was measured, and the data were fit globally to a single-site competitive binding model accounting for ligand depletion. All experiments were performed at least three times.

C2' hydrogen and Phe 366 (*Figure 3B*). In order for vcCNT to accommodate for the bulkier fluorine atom on the other epimeric position of C2', which creates steric interference and electrostatic repulsion with the π electrons of Phe 366, both Phe 366 and TM7b (including Ser 371) move slightly away from the nucleoside with respect to the uridine-bound structure (*Figure 4D*). Furthermore, the ribose of the gemcitabine is reoriented with respect to the other nucleoside-bound structures (*Figure 4E*). In addition to the steric and electrostatic disruption of the ribose-binding site, fluorine is a poor substitute for a hydroxyl as a hydrogen-bond acceptor and thus provides a less favorable interaction of the ribose with Ser 371 (*Howard et al., 1996*; *Dunitz and Taylor, 1997*). To further test the importance of the epimeric position of the 2'-hydroxyl group of the ribose, we attempted to measure the K$_D$ of another anticancer cytidine analog known as cytarabine (cytosine arabinoside) which has its 2'-hydroxyl flipped up above the ribose ring (*Figure 4B*). Cytarabine displayed no measurable binding when titrated into vcCNT (K$_D$ > 3000 µM). Interestingly, consistent with our observation with vcCNT, hCNTs show no significant binding and transport of cytarabine (*Clarke et al., 2006*). Taken together, these results reveal that the interactions of both the nucleobase and the ribose with Phe 366 and the interactions of the ribose with TM7 are critical for nucleoside recognition by CNTs, which explains the intolerance of CNTs for substituents at the other epimeric position of C2' of nucleosides.

## Structure-based ligand modification

Of all of the nucleosides and nucleoside analogs studied, the fluorescence probe pyrrolo-cytidine had the strongest binding affinity for vcCNT (K$_D$ = 0.9 µM, *Figure 5A*). We solved the crystal structure of vcCNT-7C8C bound to pyrrolo-cytidine and found that the methylpyrrole ring fits neatly into a pocket formed by TM4 and TM6 (*Figure 5B*, *Figure 5—figure supplement 1*). None of the other nucleosides in this study have moieties that can exploit this 'nucleo-pocket', and therefore this could be the root of the added strength of binding for pyrrolo-cytidine.

From our structural and binding studies, we learned that ribose interactions are important for CNT binding but nucleobase interactions are less stringent and can even be modified to improve binding. Because many nucleoside drugs contain modifications at the ribose (e.g., gemcitabine, AZT, and cytarabine), their apparent affinities for hCNTs are low (*Graham et al., 2000*; *Clarke et al., 2006*). We wondered whether this loss of affinity due to ribose modification could be compensated for by modification of the nucleobase. We synthesized a gemcitabine analog with the fluorescent nucleobase of pyrrolo-cytidine, which we now refer to as pyrrolo-gemcitabine (*Figure 5A*). We measured its binding affinity for vcCNT using the fluorescence-anisotropy assay and found that the K$_D$ decreased by ~60-fold to 24 µM, suggesting that nucleobase interactions and ribose interactions are additive.

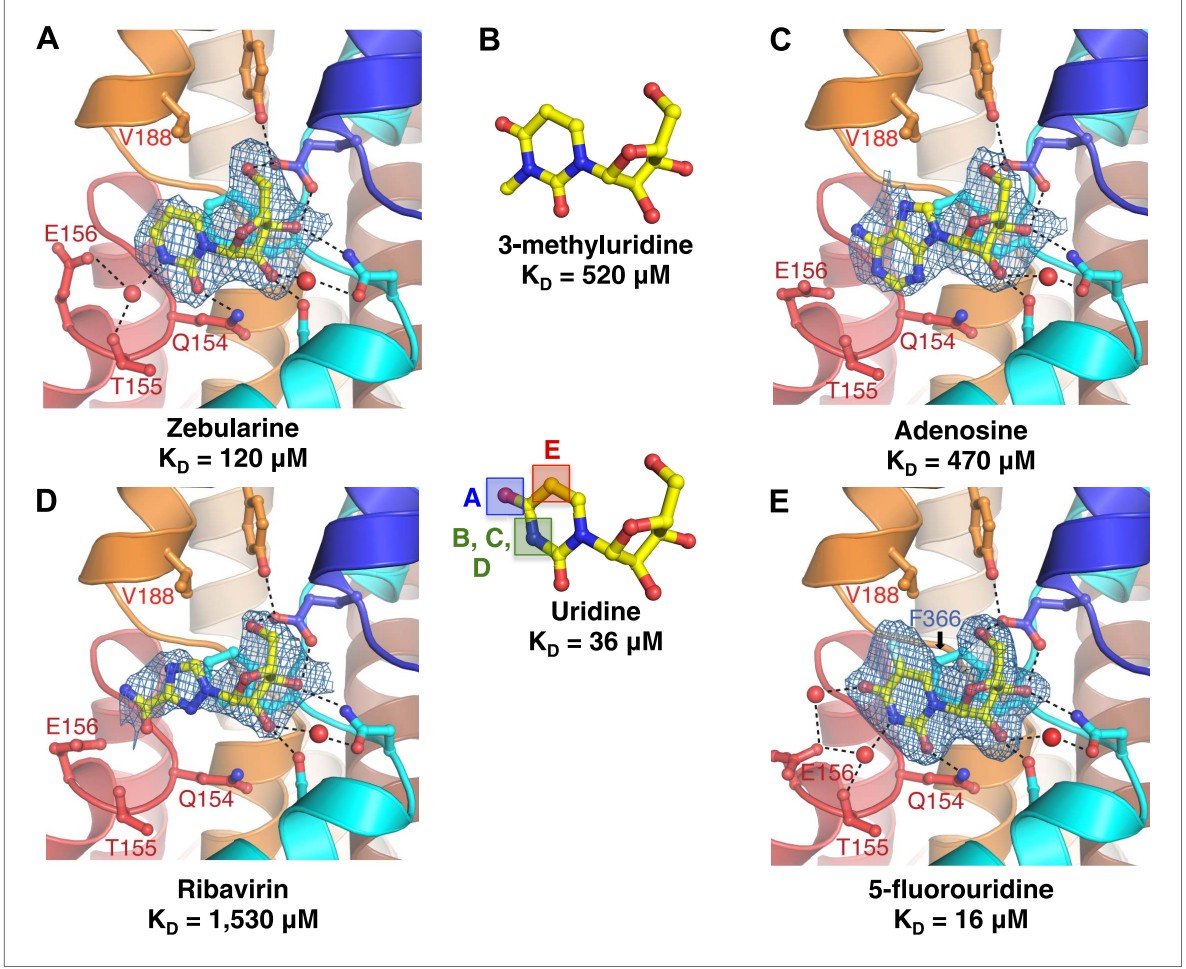

**Figure 2**. Structural basis of nucleobase recognition by vcCNT. (**A**) The crystal structure of vcCNT bound to zebularine. (**B**) Chemical structure of 3-methyluridine. (**C**) The crystal structure of vcCNT-7C8C bound to adenosine. (**D**) The crystal structure of vcCNT-7C8C bound to ribavirin. (**E**) The crystal structure of vcCNT-7C8C bound to 5-fluorouridine. Fluorine is colored cyan. All electron density maps represent $F_o$–$F_c$ SA-OMIT maps for the nucleoside contoured at 3σ. Uridine is shown in the center of the figure for reference. For stereo views of the electron density in the nucleoside-binding site for each of these structures, see **Figure 2—figure supplements 1–4**.

The following figure supplements are available for figure 2:

**Figure supplement 1**. Electron density at the nucleoside-binding site of vcCNT-zebularine.

**Figure supplement 2**. Electron density at the nucleoside-binding site of vcCNT-7C8C-adenosine.

**Figure supplement 3**. Electron density at the nucleoside-binding site of vcCNT-7C8C-ribavirin.

**Figure supplement 4**. Electron density at the nucleoside-binding site of vcCNT-7C8C-5-fluorouridine.

We next sought to test how these results translated to transportability by hCNTs. It is known that hCNT1, hCNT3, and hENT1 are the main NTs that transport gemcitabine (*Marechal et al., 2009*; *Bhutia et al., 2011*; *Damaraju et al., 2012*; *Paproski et al., 2013*). To detect nucleoside transport by hCNTs, we turned to two-electrode voltage-clamp electrophysiological recording. Because hCNTs are $Na^+$-nucleoside symporters, one can measure the current generated by the $Na^+$ transport that is coupled with nucleoside transport. We injected *Xenopus* oocytes with mRNA coding for each of the hCNTs and measured inward $Na^+$ currents elicited by the addition of different nucleosides to the extracellular side. Addition of 200 μM gemcitabine to hCNT3-expressing oocytes induced $Na^+$ currents

**Table 3.** Properties of substituents of 5-substituted uridines and their binding affinities for vcCNT

| Substituent | Radius (Å) | Electronegativity | $K_D$ (µM) |
|---|---|---|---|
| fluorine | 1.47 | 3.98 | 16 ± 1 |
| chlorine | 1.75 | 3.16 | 14 ± 4 |
| iodine | 1.98 | 2.66 | 58 ± 5 |
| methyl | 2.00 | 2.55 | 61 ± 7 |

(*Figure 5C*, *Figure 5—figure supplement 2*). However, the same amount of pyrrolo-gemcitabine had almost no effect. The lack of $Na^+$ current upon addition of pyrrolo-gemcitabine can either mean that hCNT3 is unable to bind the modified compound or hCNT3 binds but cannot transport the modified compound. To resolve this issue, a mixture of equal concentrations of both compounds was added and a reduction of $Na^+$ current was observed, suggesting that hCNT3 binds to but does not transport pyrrolo-gemcitabine. In contrast, 200 µM pyrrolo-gemcitabine elicited ~sevenfold higher total charge uptake than gemcitabine for hCNT1 (*Figure 5D*). hCNT2 transported neither compound (data not shown). Although both hCNT1 and hCNT3 can transport pyrimidines, by modifying the pyrimidine nucleobase of gemcitabine we have created a subtype-specific nucleoside analog with enhanced transportability by hCNT1.

The nucleo-pocket of vcCNT differs by only one amino acid (Gly 187 to Ser 374) from hCNT3 and three amino acids (Gly 187 to Ser 352, Val 188 to Leu 353, and Leu 259 to Val 424) from hCNT1. Since the nucleoside-binding site of vcCNT is highly homologous to hCNTs, we generated models of the hCNT1 and hCNT3 nucleo-pockets by swapping out these residues in the vcCNT-pyrrolo-cytidine structure. The structural models of hCNT1 and hCNT3 suggest changes in the overall structure of the nucleo-pocket (*Figure 6A,B*). In particular, the nucleo-pocket of the hCNT1 model does not have a large enough cavity to accommodate the pyrrole ring, and the opening to the intracellular solution is larger due to the smaller side chain on TM6. In the paradigm of the alternating-access mechanism of sodium-coupled symporters (*Krishnamurthy et al., 2009*), substrate release is achieved by the transition from the inward-facing occluded to the inward-facing open state (*Figure 6C*). Because the structure of vcCNT adopts an inward-facing occluded conformation where TM6, including Leu 259, serves as part of the gate (*Johnson et al., 2012*), the hCNT1 model suggests that the additional pyrrole group may destabilize the inward-occluded state and facilitate the transition to the inward-open state. In contrast, the additional pyrrole group may stabilize the inward-occluded state of hCNT3 and slow the transition into the inward-open state. Our hypothesis predicts that changing the structure of the nucleo-pocket in the inward-facing state would affect nucleoside transport by hCNTs. Consistent with our prediction, mutation of Val 375 and Leu 446 of hCNT3 to mimic the nucleo-pocket of hCNT1 leads to an increase in transport of pyrrolo-gemcitabine (*Figure 6D*). Furthermore, mutation of Leu 353 and Val 424 of hCNT1 to mimic the nucleo-pocket of hCNT3 leads to a decrease in transport of pyrrolo-gemcitabine (*Figure 6E*).

## Discussion

### Design principles of nucleoside recognition by CNTs

Our structural and equilibrium-binding studies of vcCNT have allowed us to better understand the design principles of nucleoside recognition by CNTs. Two helical hairpins (HP1 and HP2) and two unwound helices (TM4 and TM7), related by twofold pseudo-symmetry, create a bowl-shaped nucleoside-binding site at the center of the transport domain of vcCNT (*Figure 1B,C*). The interactions with the nucleobase are mainly formed with HP1 and TM4b, and the interactions with the ribose are mainly formed with HP2 and TM7b (*Figure 1C*). At the base of the bowl-shaped nucleoside-binding site, Phe 366 interacts with both the nucleobase and the ribose through π–π and CH–π interactions, respectively (*Figure 3A,B*). Three features make the architecture of the nucleoside-binding site of vcCNT particularly interesting: (1) the twofold pseudo-symmetry of the binding site that is divided in nucleobase and ribose binding on either side of the symmetry axis; (2) an aromatic ring at the base of the bowl that interacts with both the nucleobase and the ribose; (3) the localization of most of the protein–nucleoside interactions to one side of the nucleoside. These features of the nucleoside-binding site

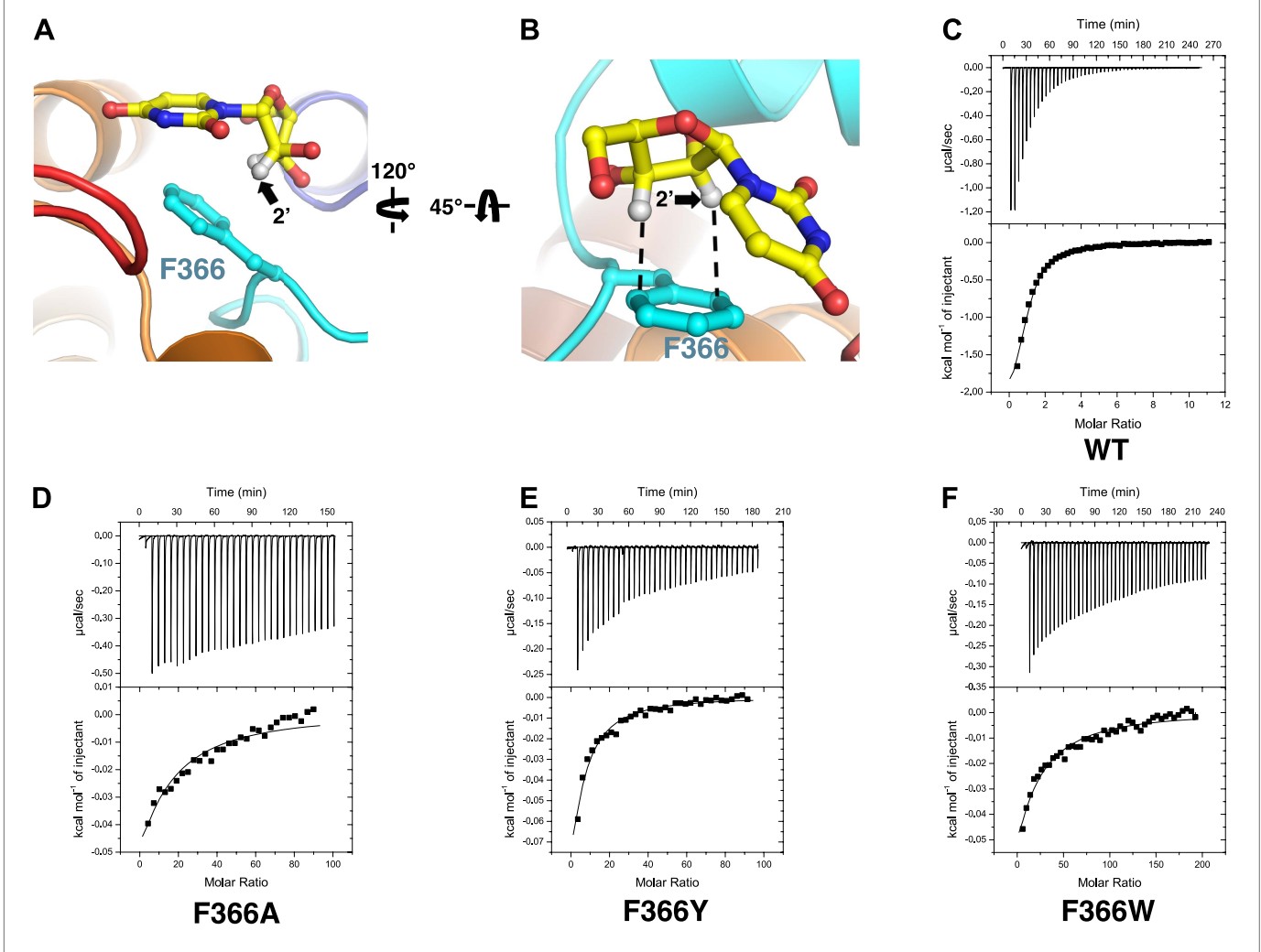

**Figure 3**. Phe 366 is crucial for nucleoside binding by vcCNT. (**A**) The nucleoside-binding site of vcCNT-7C8C bound to uridine is shown viewed from the cytoplasm. Phe 366 interacts with the uracil base via π–π interactions. The other epimeric 2' position is marked with an arrow. (**B**) Another view of the interaction between Phe 366 and uridine. Phe 366 interacts with the ribose via CH–π interactions (dashed lines). (**C**–**F**) Isothermal titration calorimetry of uridine binding to wild-type vcCNT and Phe 366 mutants. $K_D$ = 45 ± 8 μM and $\Delta H^\circ$ = −2970 ± 330 cal/mol for WT, $K_D$ = 1630 ± 120 μM and $\Delta H^\circ$ = −2200 ± 190 cal/mol for F366A, $K_D$ = 920 ± 170 μM and $\Delta H^\circ$ = −1600 ± 440 cal/mol for F366Y, and $K_D$ = 1470 ± 90 μM and $\Delta H^\circ$ = −3190 ± 130 cal/mol for F366W (means ± SEM, n = 3 measurements). Note that the $K_D$ for F366A could not be reliably measured due to the low heat associated with binding. Each of the F366 mutants is biochemically stable as evidenced by a single, sharp peak at the expected trimer size when subjected to size-exclusion chromatography (**Figure 3—figure supplement 1**).

The following figure supplement is available for figure 3:

**Figure supplement 1**. F366 mutants are biochemically stable.

lead to the following questions: What are the energetics of the nucleobase and ribose interactions? What is the role of Phe 366? Why are the interactions with the nucleoside localized to the concave side of the bowl?

With regard to binding energetics, our equilibrium-binding studies have shown that the ribose interactions are energetically important consistent with previous non-equilibrium studies with hCNTs (*Clarke et al., 2006*). However, while disruption of the nucleobase interactions can have significant effects, the nucleobase can also be modified to improve binding. It is worth noting that our equilibrium-binding studies of vcCNT translate well to hCNT function. For example, we showed that the interaction between the C3' ribose hydroxyl group and Ser 371 on TM7b is important in vcCNT

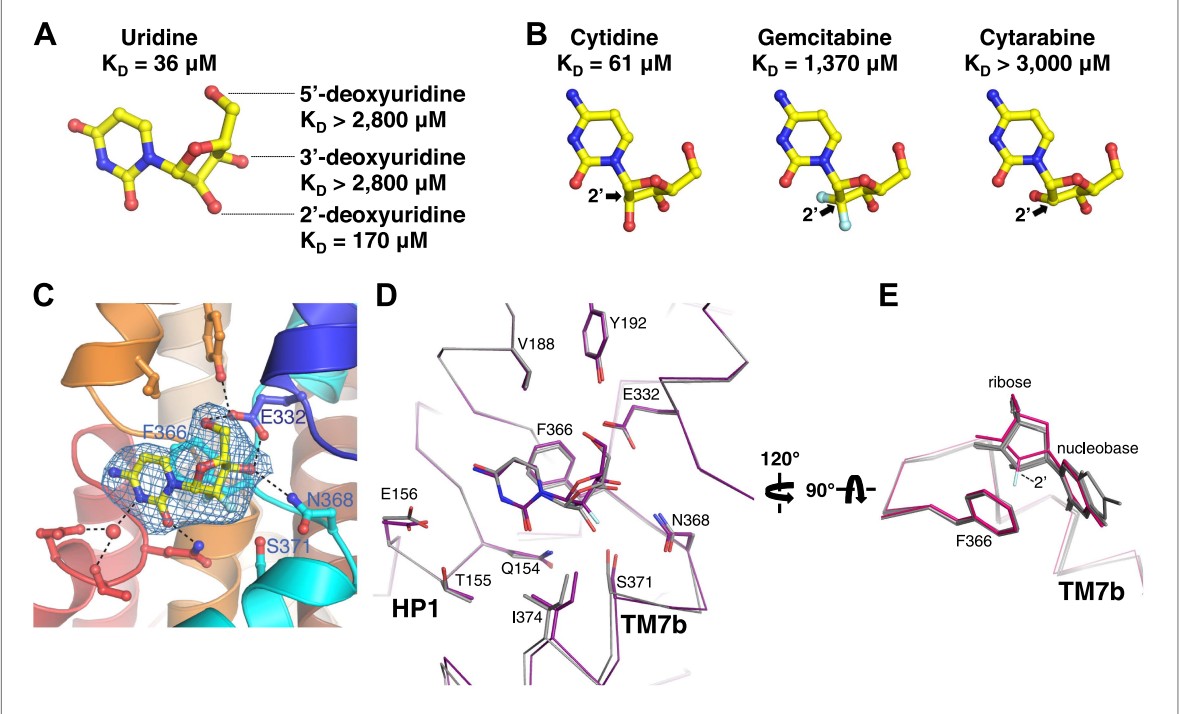

**Figure 4**. Structural basis of ribose recognition by vcCNT. (**A**) Dissociation constants for deoxyuridines. (**B**) Chemical structures and $K_D$s of cytidine, gemcitabine, and cytarabine. The cytidine is from the crystal structure of vcCNT-7C8C bound to cytidine (for a stereo view of the electron density in the nucleoside-binding site of this structure, see *Figure 4—figure supplement 1*), and the other nucleosides are simply chemical structures in the same orientation as cytidine. Fluorine atoms in gemcitabine are colored cyan. (**C**) Crystal structure of vcCNT-7C8C bound to gemcitabine. Density shown is from an $F_o$–$F_c$ SA-OMIT map contoured at 3σ. For a stereo view of the electron density in the nucleoside-binding site, see *Figure 4—figure supplement 2*. (**D**) Alignment of uridine-bound and gemcitabine-bound vcCNT structures. Structures were aligned by Cα using PyMOL. TM7 was not used for the alignment. Cα traces and interacting amino acid residues are shown. The uridine-bound vcCNT structure (PDB ID: 3TIJ) is gray and the gemcitabine-bound vcCNT-7C8C structure is deep purple. (**E**) Alignment of vcCNT and vcCNT-7C8C structures bound to uridine (PDB ID: 3TIJ), zebularine, cytidine, pyrrolo-cytidine, 5-fluorouridine, and gemcitabine. Alignments were performed in the same manner as **D**. vcCNT-7C8C-gemcitabine is shown in hot pink and all other structures are shown in gray.

The following figure supplements are available for figure 4:

**Figure supplement 1**. Electron density at the nucleoside-binding site of vcCNT-7C8C-cytidine.

**Figure supplement 2**. Electron density at the nucleoside-binding site of vcCNT-7C8C-gemcitabine.

(*Figures 1C and 4A*). Consistent with our observation, the hCNT1S546P variant is non-functional (Ser 546 in hCNT1 is equivalent to Ser 371 in vcCNT) (*Cano-Soldado et al., 2012*). Another important finding of our studies is that interactions with the nucleobase and ribose can be additive, and thus the loss of binding energy from modification of part of the nucleoside can be compensated for by the gain of energy from modification of another part of the nucleoside. Therefore, if a nucleoside drug contains a chemical modification necessary for its pharmacological function that hampers its recognition by CNTs, a compensatory chemical modification can be made so that the drug can still be recognized by CNTs.

Our structural and equilibrium-binding studies highlighted the importance of Phe 366 in nucleoside recognition. The nucleoside-bound structures of vcCNT help to shed light upon the structural basis of the importance of this residue. Notably, Phe 366 is the only residue within the nucleoside-binding site that forms interactions with both portions of the nucleoside. Furthermore, all of the other binding-site residues on HP1, HP2, TM4b, and TM7b interact from the same side of the nucleoside as Phe 366, forming the shape of a bowl around the nucleoside with Phe 366 serving as its base. As a result, the nucleoside rests on the face of Phe 366, likely helping to orient the nucleoside so that it may form all of the other

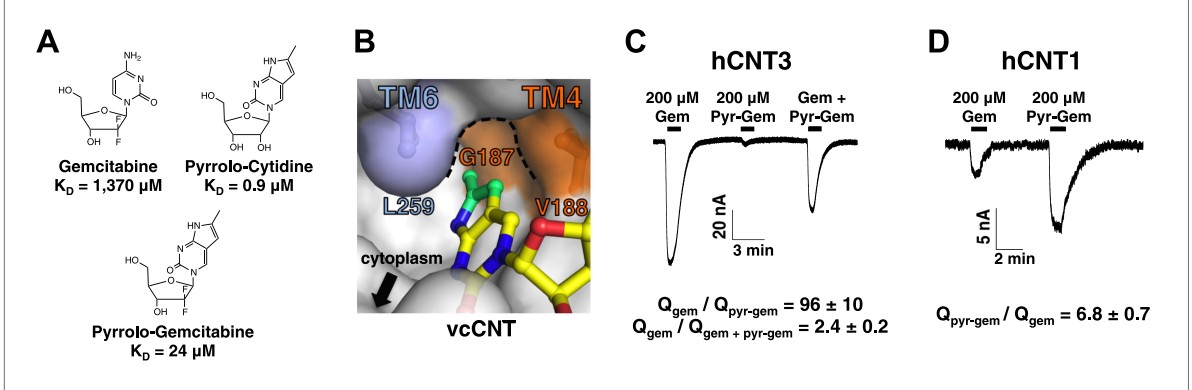

**Figure 5**. Design of pyrrolo-gemcitabine and its transportability by hCNTs. (**A**) Chemical structures and $K_D$s of gemcitabine and the pyrrolo-nucleosides. (**B**) The crystal structure of vcCNT-7C8C in complex with pyrrolo-cytidine is shown in surface representation with pyrrolo-cytidine shown in stick representation. The additional three carbons that comprise the methylpyrrole ring of pyrrolo-cytidine are colored green. The vcCNT-7C8C nucleo-pocket, formed mainly by G187 (TM4), V188 (TM4), and L259 (TM6), is delineated with a dotted line. The location of the cytoplasm, adjacent to the nucleo-pocket, is shown. For a stereo view of the electron density in the nucleoside-binding site, see ***Figure 5—figure supplement 1***. (**C**) hCNT3 transports gemcitabine but not pyrrolo-gemcitabine. $Na^+$ currents were elicited by the addition of nucleoside to *Xenopus* oocytes expressing hCNT3, and currents were measured by two-electrode voltage-clamp. An example current trace is shown. For each individual oocyte, the area under each current peak was measured to calculate total charge (Q) transported during application of nucleoside. The ratio of total charge co-transported with gemcitabine to that with pyrrolo-gemcitabine or gemcitabine total charge to gemcitabine + pyrrolo-gemcitabine total charge was calculated for each oocyte experiment (means ± SEM, n = 14 oocytes). For Gem + Pyr-Gem, 200 µM of each nucleoside was added simultaneously. (**D**) hCNT1 transports pyrrolo-gemcitabine better than gemcitabine. Same experiment as in **C** but hCNT1-expressing oocytes were used and the ratio of total charge for pyrrolo-gemcitabine to gemcitabine is shown (means ± SEM, n = 11 oocytes). Neither gemcitabine nor pyrrolo-gemcitabine elicited currents in water-injected oocytes (***Figure 5—figure supplement 2***) See ***Figure 5—source data 1*** for total charge source data.

The following source data and figure supplements are available for figure 5:

**Source data 1**. Total charge data.

**Figure supplement 1**. Electron density at the nucleoside-binding site of vcCNT-7C8C-pyrrolo-cytidine.

**Figure supplement 2**. Water-injected oocytes do not respond to gemcitabine or pyrrolo-gemcitabine treatment.

interactions within the binding site. Although π–π and CH–π interactions are generally weak, they can provide significant interaction energies depending on the circumstance (***Waters, 2002***; ***Nishio, 2011***). The importance of Phe 366 is demonstrated by the changes in affinity of C5-substituted uridine analogs for vcCNT by altering the π–π interaction with the nucleobase and the significant reduction of affinity of C2'-substituted nucleoside drugs via disruption of the CH–π interaction with the ribose. Furthermore, ITC experiments with Phe 366 mutants revealed the stringent requirement for phenylalanine at this position, as even replacement with tyrosine resulted in a significant loss of binding affinity for uridine. Taken together, we propose that the role of Phe 366 is to position the nucleoside for effective binding thus serving as a 'selectivity ring'.

What is the molecular basis of having a bowl-shaped nucleoside-binding site? Several other nucleoside-binding proteins bind to their substrates by sandwiching the nucleobase between aromatic residues (***Suzuki et al., 2004***; ***Monecke et al., 2014***). The utilization of a bowl-shaped binding site for a transporter makes practical sense as the substrate must be able to bind to and dissociate readily from the binding site in order for efficient transport to occur. We previously proposed that the transport domain undergoes a rigid-body motion while the scaffold domain, including TM6, remains static during the transition from the outward-occluded to the inward-occluded conformational state (***Figure 6C***; ***Johnson et al., 2012***). In the inward-occluded conformation, TM6 (including Leu 259) is located on top of the bowl, serving as part of the intracellular gate and partially occluding the nucleoside from dissociating into the cytoplasm. Portions of HP1 and TM7b form the rest of the intracellular gate and may move slightly during the transition from the inward-occluded to the inward-open conformational state (***Figure 6C***), allowing the nucleoside to exit from the top of the bowl.

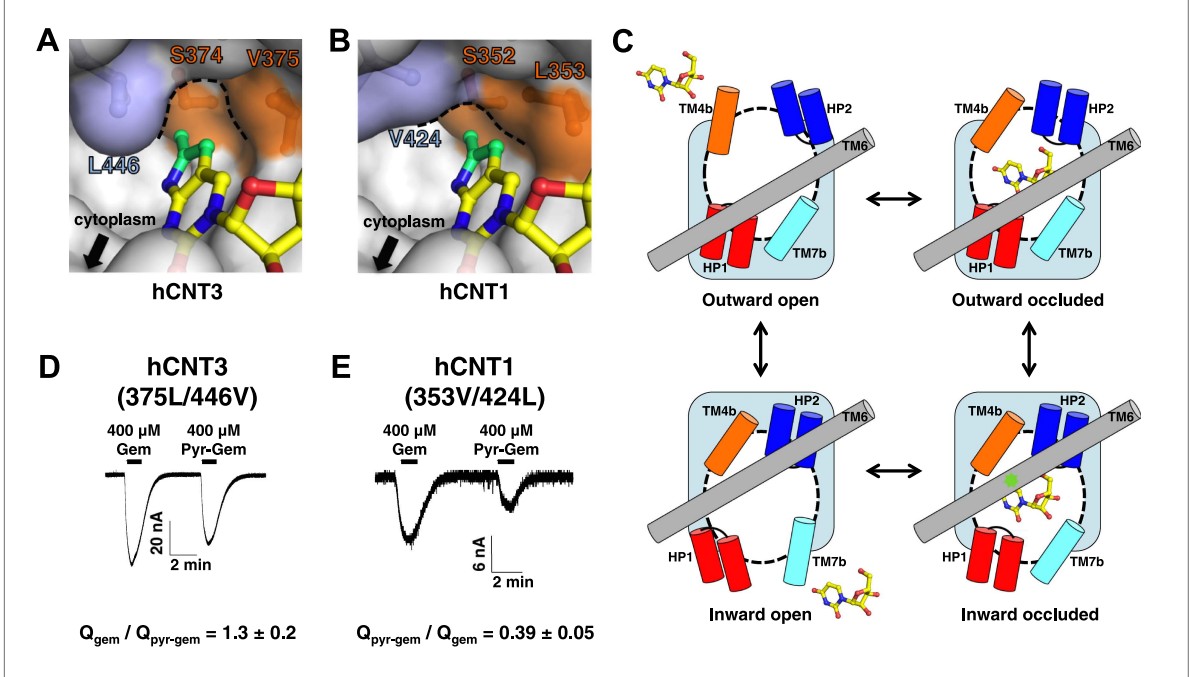

**Figure 6**. Structural basis of the subtype selectivity of pyrrolo-gemcitabine. (**A**) Model of hCNT3 nucleo-pocket. The structure of vcCNT-7C8C bound to pyrrolo-cytidine was used to generate a model of the hCNT3 nucleo-pocket by mutating the appropriate residues in PyMOL and selecting the rotamer that yielded the lowest amount of steric clash. (**B**) Model of hCNT1 nucleo-pocket. The model was generated in the same manner as **A**. Note that the methyl-pyrrole ring (green) will clash with the hCNT1 nucleo-pocket if it maintains the nucleoside-binding mode observed in the vcCNT structure. (**C**) Hypothetical alternating-access mechanism of vcCNT. A cartoon representation of the different conformational states along the transport cycle is depicted. The transport domain (including HP1, HP2, TM4b, and TM7b) and TM6 are shown as cylinders. Uridine is shown in stick representation. The nucleo-pocket in the inward-occluded conformation (bottom right) is located between TM6 and TM4 and is marked with a green star. The inward-occluded conformation is derived from the crystal structures of vcCNT. All other conformations are purely hypothetical. The transition between inward- and outward-facing conformations has been proposed to be achieved by a rigid-body movement of the transport domain across TM6 (***Johnson et al., 2012***). Extracellular and intracellular gating likely involves slight rearrangements of HP2/TM4b and HP1/TM7b, respectively. (**D**) hCNT3 (375L/446V) is capable of transporting both gemcitabine and pyrrolo-gemcitabine. Same experiment as ***Figure 5C,D*** but hCNT3 (375L/446V)-expressing oocytes were used and the ratio of total charge (Q) co-transported with gemcitabine to pyrrolo-gemcitabine is shown (means ± SEM, n = 11 oocytes). (**E**) hCNT1 (353V/424L) transports pyrrolo-gemcitabine less efficiently than gemcitabine (means ± SEM, n = 8 oocytes). Note that a higher nucleoside concentration was needed for **D** and **E** than the wild-type experiments due to lower transporter activity and/or expression. See ***Figure 6—source data 1*** for total charge source data.

The following source data is available for figure 6:

**Source data 1**. Total charge data.

This rigid-body conformational change also suggests that most of the interactions between transporter and nucleoside that are present in the inward-occluded structure are maintained in the outward-facing conformation, as observed with other sodium-coupled transporters (***Reyes et al., 2009***; ***Zhou et al., 2014***). One exception is the interaction between the methylpyrrole ring of pyrrolo-cytidine with the nucleo-pocket, which is likely to preferentially interact with the inward-occluded conformation because of the involvement of part of the scaffold domain (TM6, ***Figures 5B and 6C***), which we expect to be immobile during the conformational change. While both conformations should be represented when the transporter is solubilized in detergent micelles, we anticipate that our structural and equilibrium-binding data revealed most of the amino acid residues that are important for nucleoside recognition. Consistent with our hypothesis, amino acid residues that were shown to be important from our inward-conformation-based binding studies in detergent have also been shown to be important for the binding and transport of nucleosides by human CNTs in several cell-based mutational studies (***Loewen et al., 1999***; ***Zhang et al., 2003***, ***2005***; ***Yao et al., 2007***; ***Slugoski et al., 2009***). Further structural and biophysical studies probing the outward-facing state will help us to develop a complete understanding of the principles of nucleoside recognition by CNTs.

## Structure-based ligand modification and its implications

Despite the recent progress in the area of structural biology of transporters, most SLC transporter structures determined to date are not highly homologous to human transporters, and it is often the case that drug interactions with these non-human transporters are different from their human counterparts, rendering structural studies of drug–transporter interactions technically challenging and necessitating substantial engineering to mimic the behavior of the human transporters (*Singh et al., 2007*; *Wang et al., 2013*). Although our CNT is prokaryotic in origin, it is an excellent model system to study human CNTs and offers us the opportunity to conduct structural studies of nucleoside and nucleoside-drug selectivity by hCNTs.

Understanding the structural principles of nucleoside and nucleoside drug recognition by vcCNT not only allowed us to understand why a certain class of drugs (e.g., gemcitabine and cytarabine) are not well recognized and transported by hCNTs, but also offered us an opportunity to modify an existing drug to improve its affinity for vcCNT. Furthermore, our electrophysiological studies show that it is not only transported more efficiently by hCNT1, but it is also selectively transported by hCNT1. Our structural models of hCNT1 and hCNT3 allowed us to hypothesize that the subtype-specific differences in the structures of the nucleo-pocket in the inward-facing-occluded conformation give rise to the subtype-selectivity of the modified compound. The results of the mutational studies of the nucleo-pockets in hCNTs are consistent with this hypothesis. Although we do not know whether the modification of gemcitabine affects the outward conformation due to the lack of a structure of the outward-facing conformation, our studies suggest that destabilization of the inward-facing-occluded step would facilitate the release of the pyrrolo-gemcitabine. Conversely, stabilization of the outward-facing-occluded step would facilitate the capture of the substrate. Taken together, if one has knowledge of both the outward- and inward-facing conformations of the transporter, it might be possible, at least in principle, to modify a compound to bind to both conformations with differential affinities, which may improve its transportability and selectivity. Prior to our studies, the presence of the nucleo-pocket structure was unknown. Although our study with pyrrolo-gemcitabine serves merely as proof of concept, it is conceivable that the nucleo-pocket structure can be utilized in the design of nucleoside-derived drugs or prodrugs that can be specifically targeted only to cell types that express hCNT1 since expression levels of hCNT1 are closely related to the responsiveness of many different types of normal or cancer cells to chemotherapy treatment (*Lane et al., 2010*; *Naito et al., 2010*; *Rabascio et al., 2010*; *Bhutia et al., 2011*; *Choi, 2012*).

Finally, our studies provide another valuable concept: even though two transporter subtypes may share substrate and drug specificity, as is the case with hCNT1 and hCNT3, it is still possible to use structural differences (with the help of modeling) between the two subtypes to design or modify a drug that can be selectively transported. This concept has broad applications to many SLC transporters that are involved in ADME since many families such as SLC21, SLC22, and SLC29 possess subtype-dependent drug specificities and/or tissue distributions (*Baldwin et al., 2004*; *Koepsell and Endou, 2004*; *Hagenbuch and Stieger, 2013*).

These results can also have an impact on basic scientific research. Since some transporter subtypes show significant changes in expression levels between normal and pathological conditions (i.e., cancer), and these changes in transporter expression are usually important for the pathological conditions to persist, a fluorescent compound that is subtype-specific (e.g., pyrrolo-gemcitabine) for a certain transporter family can be a valuable tool to study the role of transporter subtypes in human health and disease through live cell imaging (*Farre et al., 2004*; *Zhang et al., 2006*; *Bhutia et al., 2011*; *Perez-Torras et al., 2013*). Taken together, this work not only represents an structural study of substrate and drug selectivity by membrane transporters, but our results also provide proof of principle for using this type of structure-function study for modifying drugs so that they are recognized and taken up into the cell by their cognate transporters more efficiently and selectively (*Han and Amidon, 2000*; *Majumdar et al., 2004*).

# Materials and methods

## Crystallization

Wild-type vcCNT and vcCNT-7C8C were expressed and purified as described (*Johnson et al., 2012*) in the absence of any added nucleoside. Briefly, protein was expressed as a His$_{10}$-MBP fusion in C41

(DE3) cells, cells were lysed by homogenizer (AVESTIN, Ottawa, ON), protein was extracted from crude lysate using 30 mM dodecyl-maltoside, lysates were spun down to remove the insoluble fraction, and the supernatant was applied to a $Co^{2+}$-affinity column for purification. The $His_{10}$-MBP was cleaved by overnight digestion by PreScission Protease, and vcCNT was separated from $His_{10}$-MBP by gel filtration using a Superdex 200 10/300 GL column in the presence of 5 mM decyl-maltoside. After gel filtration, protein was concentrated to ~10 mg/ml and nucleoside was added to 1 mM (uridine), 2 mM (cytidine, adenosine), or 10 mM (ribavirin, gemcitabine, 5-fluorouridine, pyrrolo-cytidine, zebularine). Crystals were grown in the presence of 100 mM $CaCl_2$, 37–42% PEG400, and 100 mM buffer: HEPES pH 7.5 (ribavirin), Tris–HCl pH 8.0–8.5 (adenosine, cytidine, gemcitabine, pyrrolo-cytidine), or glycine pH 9.5 (uridine, 5-fluorouridine, zebularine). Crystals were grown using the microbatch-under-oil technique. Crystals were harvested after 10–14 days, transferred to cryo solution containing 32.5% PEG400, and flash frozen in liquid nitrogen.

## Data collection and structure determination

X-ray data were collected at beamlines 22-ID-D and 24-ID-C at the Advanced Photon Source at Argonne National Laboratory. Data were processed using HKL-2000. The uridine, cytidine, zebularine, 5-fluorouridine, ribavirin, and pyrrolo-cytidine complex structures were refined using PHENIX with the original vcCNT structure (PDB ID 3TIJ) as the input model. The adenosine and gemcitabine complex structures were solved by molecular replacement with the original vcCNT structure as the search model using PHASER and refined using PHENIX.

## Fluorescence-based equilibrium-binding assay

To measure the binding affinity of vcCNT for fluorescent nucleoside analogs, individual 500-µl solutions were prepared containing varying concentrations of vcCNT in 5 mM DM and either 5 µM pyrrolo-cytidine or 2 µM pyrrolo-gemcitabine. The fluorescence anisotropy of each solution at $\lambda_{ex}$ = 340 nm and $\lambda_{em}$ = 467 nm was measured using a Cary Eclipse Fluorescence Spectrophotometer (Agilent, Santa Clara, CA) with automated polarizers. Each titration was performed at least three times. The data for the three titrations were simultaneously fit to a single-site binding model based off of Morrison's quadratic equation using nonlinear least-squares analysis in *GraphPad Prism* to obtain a dissociation constant and standard error.

To measure the binding affinity of vcCNT for other nucleosides and nucleoside analogs, the nucleoside of interest was titrated 5 µl at a time into 500 µl of solution initially containing 5 µM of vcCNT in 5 mM DM and 1–2 µM pyrrolo-cytidine. The fluorescence anisotropy after each addition was measured. Each titration was performed at least three times. The data for the three titrations were simultaneously fit to a one-site competitive binding model based off of Wang's method (*Wang, 1995*) using nonlinear least-squares analysis in *GraphPad Prism* to obtain a dissociation constant and standard error.

## Isothermal titration calorimetry

vcCNT mutants were prepared in the same manner as wild-type vcCNT. 15–30 mM of uridine was titrated 5 µl at a time into 25–40 µM of vcCNT solubilized in 5 mM DM using a MicroCal VP-ITC system (GE Healthcare, Pittsburgh, PA). The total heat exchanged during each injection was fit to a single-site binding isotherm with $K_D$ and $\Delta H^\circ$ as independent parameters.

## Chemical synthesis

See *Supplementary file 1* for a full description of the pyrrolo-gemcitabine synthesis. Briefly, the Sonogashira coupling of known 2′-deoxy-2′,2′-difluoro-5-iodo-uridine (*Quintiliani et al., 2011*) with propyne followed by Cu(I)-mediated cyclization provided the corresponding furano-gemcitabine in 64% for 2 steps. Treatment of furano-gemcitabine with $NH_4OH$ and $CH_3OH$ completed the synthesis of the desired pyrrolo-gemcitabine in 75%.

## *Xenopus laevis* oocyte expression and electrophysiology

The genes coding for the three hCNTs and vcCNT were cloned into a pGEM-HE vector. Plasmids were linearized using either SphI or NheI, and mRNA was transcribed using the mMESSAGE mMACHINE T7 Transcription Kit (Ambion, Grand Island, NY). Defolliculated *Xenopus laevis* oocytes were purchased from Ecocyte Bioscience (Austin, TX). Individual oocytes were injected with 40 ng of mRNA using a 10-µl microdispenser (Drummond Scientific, Broomall, PA) fitted with a tapered

glass pipette tip and incubated at 17 °C for 4–5 days in ND96 buffer (96 mM NaCl, 2 mM KCl, 1 mM MgCl$_2$, 1.8 mM CaCl$_2$, and 5 mM HEPES pH 7.5) with 0.1% penicillin and streptomycin before recording.

The oocyte-recording chamber was gravity-perfused with ND96 buffer at a rate of 2 ml/min. Membrane currents were measured using an Oocyte Clamp (OC-725C; Warner Instruments, Hamden, CT). Individual oocytes were penetrated with two microelectrodes filled with 3 M KCl (0.5–1.0 MΩ). All electrophysiological experiments were conducted at room temperature. The OC-725C Oocyte Clamp was computer-interfaced via an Axon Digidata 1550 and controlled by Axoscope software (Molecular Devices, Sunnyvale, CA). The current signals were filtered at 20 Hz and sampled at intervals of 20 ms. The signals were filtered at 0.5 Hz by use of pCLAMP 10.4 software for data presentation. Ooctyes were impaled with the electrode filled with 3 M KCl, and then membrane potentials were observed for 10 min. Cells were discarded if resting membrane potentials were unstable or more positive than −30 mV. Oocyte membrane potentials were clamped at −90 mV for holding potentials to measure transporter-generated currents. All data are shown as means ± SEM.

## Acknowledgements

Data for this study were collected at beamlines SER-CAT 22-ID-D and NE-CAT 24-ID-C at the Advanced Photon Source. We thank K Daniels for helping to develop the fluorescence assay and aiding in the initial analysis of binding data, and F Valiyaveetil and S Lockless for critical reading. Initial X-ray screening of crystals was performed at the Duke macromolecular crystallography facility. This work was supported by NIH R01 GM100984 (S-YL) and Duke Cancer Institute Pilot funding (S-YL and JH). S-YL is a McKnight Scholar, Klingenstein fellow, Alfred P Sloan Research fellow, Mallinckrodt Scholar, Whitehead Scholar, Basil O'Connor Starter Scholar, and NIH Director's New Innovator awardee.

Author Information

Atomic coordinates and structure factors for the reported crystal structure are deposited in the Protein Data Bank under accession codes 4PB1 (ribavirin), 4PB2 (5-fluorouridine), 4PD5 (gemcitabine), 4PD6 (uridine), 4PD7 (zebularine), 4PD8 (pyrrolo-cytidine), 4PD9 (adenosine), and 4PDA (cytidine).

## Additional information

### Funding

| Funder | Grant reference number | Author |
|---|---|---|
| National Institute of General Medical Sciences | R01GM100984 | Zachary Lee Johnson, Jun-Ho Lee, Seok-Yong Lee |

The funder had no role in study design, data collection and interpretation, or the decision to submit the work for publication.

### Author contributions

ZLJ, Conception and design, Acquisition of data (structure determination, fluorescence and ITC experiments, and radioactive flux assays), Analysis and interpretation of data, Wrote the manuscript; J-HL, Acquisition of data (TEVC experiments); KL, ML, D-YK, Acquisition of data (chemical synthesis); JH, Conception and design; S-YL, Conception and design, Analysis and interpretation of data, Wrote the manuscript

## Additional files

### Supplementary file

• Supplementary file 1. Synthetic approach to furano- and pyrrolo-gemcitabines.

## Major datasets

The following datasets were generated:

| Author(s) | Year | Dataset title | Dataset ID and/or URL | Database, license, and accessibility information |
|---|---|---|---|---|
| Johnson ZL, Lee SY | 2014 | Crystal structure of vcCNT-7C8C bound to ribavirin | http://www.rcsb.org/pdb/explore/explore.do?structureId=4PB1 | Publicly available at RCSB Protein Data Bank. |
| Johnson ZL, Lee SY | 2014 | Crystal structure of vcCNT-7C8C bound to 5-fluorouridine | http://www.rcsb.org/pdb/explore/explore.do?structureId=4PB2 | Publicly available at RCSB Protein Data Bank. |
| Johnson ZL, Lee SY | 2014 | Crystal structure of vcCNT-7C8C bound to gemcitabine | http://www.rcsb.org/pdb/explore/explore.do?structureId=4PD5 | Publicly available at RCSB Protein Data Bank. |
| Johnson ZL, Lee SY | 2014 | Crystal structure of vcCNT-7C8C bound to uridine | http://www.rcsb.org/pdb/explore/explore.do?structureId=4PD6 | Publicly available at RCSB Protein Data Bank. |
| Johnson ZL, Lee SY | 2014 | Crystal structure of vcCNT bound to zebularine | http://www.rcsb.org/pdb/explore/explore.do?structureId=4PD7 | Publicly available at RCSB Protein Data Bank. |
| Johnson ZL, Lee SY | 2014 | Crystal structure of vcCNT-7C8C bound to pyrrolo-cytidine | http://www.rcsb.org/pdb/explore/explore.do?structureId=4PD8 | Publicly available at RCSB Protein Data Bank. |
| Johnson ZL, Lee SY | 2014 | Crystal structure of vcCNT-7C8C bound to adenosine | http://www.rcsb.org/pdb/explore/explore.do?structureId=4PD9 | Publicly available at RCSB Protein Data Bank. |
| Johnson ZL, Lee SY | 2014 | Crystal structure of vcCNT-7C8C bound to cytidine | http://www.rcsb.org/pdb/explore/explore.do?structureId=4PDA | Publicly available at RCSB Protein Data Bank. |

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
