## [Decision Letter]

Thank you for sending your work entitled “Structural basis of nucleoside and nucleoside drug selectivity by concentrative nucleoside transporters” for consideration at *eLife*. Your article has been favorably evaluated by Michael Marletta (Senior editor) and 2 reviewers, one of whom is a member of our Board of Reviewing Editors.

The following individuals responsible for the peer review of your submission have agreed to reveal their identity: Volker Dötsch, Raimund Dutzler.

The Reviewing editor and the other reviewer discussed their comments before we reached this decision, and the Reviewing editor has assembled the following comments to help you prepare a revised submission.

The authors describe a thorough investigation of the interaction of vcCNT with different substrate analogues, including important antiviral and anticancer drugs. vcCNT is a close prokaryotic homologue of the SLC28 family of Na-coupled nucleoside transporters, that are responsible for the import of nucleoside analogues into cells. Its structure was previously determined at high resolution by the same group. The binding mode of ligand analogues to an inward-facing conformation of the protein was investigated by X-ray crystallography. The data is of high quality and of sufficient resolution to analyze interactions. Next to the crystallographic studies, the binding affinity to the detergent solubilized protein was quantified by a competition assay with a fluorescent ligand and by isothermal titration calorimetry. Taken together the data provides detailed insight into the structural and energetic determinants of the interaction.

A few points, however, should be addressed before publication:

1) To be able to judge the crystallographic data it would be helpful if the authors could provide stereo figures of the different complexes with superimposed 2Fo-Fc electron density in the supplemental materials.

2) For the binding studies it is not clear whether the authors expect to measure binding to an inward-facing conformation, or whether they believe that the transporter may adopt different conformations in solution.

3) Figures 5 and 6, E show bars of the average of the integrated charge from 10-12 independent experiments. It is not clear how the authors have accounted for differences in the transport currents due to different expression levels. For this figure, it would also be necessary to show the corresponding traces from non-injected oocytes as control.

4) Is the change in subtype specificity from hCNT3 to hCNT1 important for the delivery of this class of compounds, or was this study designed as proof of concept?

5) The results of the mutational analysis of F366 could potentially also be explained with misfolding of the protein. Has the integrity of the mutated transporter been investigated?

---

## [Author Response]

*1) To be able to judge the crystallographic data it would be helpful if the authors could provide stereo figures of the different complexes with superimposed 2Fo-Fc electron density in the supplemental materials*.

We have now provided supplemental stereo figures of the nucleoside-binding site of vcCNT in complex with each of the nucleoside analogs with 2Fo-Fc electron density maps superimposed.

*2) For the binding studies it is not clear whether the authors expect to measure binding to an inward-facing conformation, or whether they believe that the transporter may adopt different conformations in solution*.

vcCNT, although it is a novel fold, adopts a common overall architecture of a static scaffold domain and a mobile transport domain, which is shared by several sodium-coupled transporters (e.g. Glt_Ph_). On the basis of our vcCNT structure and structural and biophysical studies of related transporters, we previously hypothesized that transport is achieved by a rigid-body motion of the transport domain that maintains similar structures for both the inward- and outward-occluded conformations, and thus the interactions with the nucleoside are similar in both conformations. Our hypothesis is in line with recent studies done by the Boudker group that show that Glt_Ph_ samples both conformations with equal probability and the affinities of both conformations are similar in detergent micelles. Thus we assume that most interactions that we showed to be important using nucleoside analogs have effects in both conformations. We think that one exception is the interaction between the methylpyrrole ring of pyrrolo-cytidine with the nucleopocket, which must affect the inward-occluded conformation preferentially because of the involvement of part of the static scaffold domain (TM6). Consistent with our assumption, several mutational studies done on human CNTs show that most of the residues that we believe are important in the inward-facing conformation are also important for the binding and transport of nucleosides. Further structural and biophysical studies of the outward-occluded conformation and associated conformational dynamics will help test our hypothesis.

We have added the following paragraph to the Discussion section: “This rigid-body conformational change also suggests that most of the interactions between transporter and nucleoside that are present in the inward-occluded structure are maintained in the outward-facing conformation, as observed with other sodium-coupled transporters (45, 60). One exception is the interaction between the methylpyrrole ring of pyrrolo-cytidine with the nucleopocket, which is likely to preferentially interact with the inward-occluded conformation because of the involvement of part of the scaffold domain (TM6, Figures 5 and 6), which we expect to be immobile during the conformational change. While both conformations should be represented when the transporter is solubilized in detergent micelles, we anticipate that our structural and equilibrium-binding data revealed most of the amino acid residues that are important for nucleoside recognition. Consistent with our hypothesis, amino acid residues that were shown to be important from our inward-conformation-based binding studies in detergent have also been shown to be important for the binding and transport of nucleosides by human CNTs in several cell-based mutational studies. Further structural and biophysical studies probing the outward-facing state will help us to develop a complete understanding of the principles of nucleoside recognition by CNTs.”

*3)*
Figures 5 and 6
*show bars of the average of the integrated charge from 10-12 independent experiments. It is not clear how the authors have accounted for differences in the transport currents due to different expression levels. For this figure, it would also be necessary to show the corresponding traces from non-injected oocytes as control*.

To accurately measure differences in the compound-dependent Na^+^ transport within the same oocyte, we collected data from oocytes displaying high levels of currents, which could be the reason for the small deviations in the total charge transported from different oocytes. The control water-injected oocytes showed no response to either gemcitabine or pyrrolo-gemcitabine. We have now included the corresponding current traces from the control water-injected oocytes in Figure 5—figure supplement 2. We have also replaced the bar graphs in Figures 5 and 6 with the ratios of total charge transported upon gemcitabine and pyrrolo-gemcitabine treatment. The ratios were calculated for each individual oocyte experiment, and the values displayed in the figures are the averaged ratios from several different oocyte experiments.

*4) Is the change in subtype specificity from hCNT3 to hCNT1 important for the delivery of this class of compounds*, *or was this study designed as proof of concept?*

Our study was solely designed as proof of concept. We just want to point out that at least in principle this type of specificity switch could be utilized to target cancer cells that express higher levels of hCNT1 while having little effect on those expressing higher levels of hCNT3. There are multiple studies that have shown that several gemcitabine-resistant pancreatic cancer cells reduce their hCNT1 surface expression levels, suggesting that hCNT1 is the main gemcitabine transporter for these cancer cell types. Conversely, there is a clinical study showing that expression levels of hCNT3 are important for the survival of pancreatic cancer patients with a specific type of gemcitabine treatment, suggesting that an hCNT3-specific gemcitabine derivative could also be effective in this specific case.

We have modified a paragraph in the Discussion section to clarify that our studies with pyrrolo-gemcitabine are designed as proof of concept: “Although our study with pyrrolo-gemcitabine serves merely as proof of concept, it is conceivable that the nucleo-pocket structure can be utilized in the design of nucleoside-derived drugs or prodrugs that can be specifically targeted only to cell types that express hCNT1 since expression levels of hCNT1 are closely related to the responsiveness of many different types of normal or cancer cells to chemotherapy treatment.”

*5) The results of the mutational analysis of F366 could potentially also be explained with misfolding of the protein*. *Has the integrity of the mutated transporter been investigated?*

The F366 mutants (F366A, F366Y, and F366W) behave similarly to the wild-type transporter in terms of their biochemical stability and elution volumes in size-exclusion chromatography. Because we demonstrated in our initial structure paper that wild-type vcCNT exists as a trimer both in detergent micelles and in the membrane, the similar elution volumes of the mutants strongly suggest that the mutants adopt trimer stoichiometry as well. Furthermore, we did not observe any aggregation of the mutants during our ITC experiments. We have now included gel filtration traces of the mutants as well as wild-type vcCNT in Figure 3—figure supplement 1.